# Disproportionality analysis of fondaparinux associated adverse events based on the FDA adverse event reporting system

Fengjiao Kang[1,2], Yin Wang[1], Fengqun Cai[1], Liuyun Wu[1], Lizhu Han[1], Qinan Yin[1], Yuan Bian[1]*

1 Department of Pharmacy, Personalized Drug Research and Therapy Key Laboratory of Sichuan Province, Sichuan Provincial People's Hospital, School of Medicine, University of Electronic Science and Technology of China, Chengdu, China, 2 Pharmacy Department of Xinjiang Medical University Affiliated Traditional Chinese Medicine Hospital, Urumqi Xinjiang, China

* bianyuan567@126.com

## Abstract

Fondaparinux is a widely used anticoagulant for treating venous thromboembolism and acute coronary syndrome by inhibiting factor Xa. However, it carries a risk of bleeding. This study analyzes its safety using the FDA Adverse Event Reporting System (FAERS) database to guide clinical use and future research. Data from 2004 Q1 to 2024 Q3 were examined using Reporting Odds Ratio (ROR), Proportional Reporting Ratio (PRR), Bayesian Confidence Propagation Neural Network (BCPNN), and Empirical Bayesian Geometric Mean (EBGM) methods. Among 21,483,491 reports, 5,788 were related to fondaparinux, with 17,523 adverse events (AEs). The most affected systems were vascular (n = 2046) and blood/lymphatic (n = 1125). Abnormal coagulation factor X concentration, thrombosis with thrombocytopenia syndrome, and incision site hematoma had the strongest signals. AEs were more frequent in females and elderly patients, especially within the first month. Additional Preferred Term (PT)–level signals defined in MedDRA were observed, including heparin induced thrombocytopenia and eosinophilia. Heightened pharmacovigilance particularly in older adults may be warranted. This study enhances understanding of fondaparinux's safety, providing insights for reducing risks and ensuring safer clinical application.

## 1. Introduction

Fondaparinux, a synthetic pentasaccharide anticoagulant, is widely used in clinical practice for the prevention and treatment of venous thromboembolic diseases, acute coronary syndrome, and other thrombotic disorders [1–5]. It exerts its anticoagulant activity by binding to antithrombin III (AT-III) with high selectivity [6], thereby enhancing AT III–mediated inhibition of coagulation factor Xa (FXa) by nearly 300 fold [7,8]. FXa represents a pivotal junction in the coagulation cascade, where both intrinsic

**Data availability statement:** All data is publicly available on the FDA website (https://fis.fda.gov/extensions/FPD-QDE FAERS/FPD-QDE-FAERS.html). The data and code supporting the findings of this study are available in the Figshare repository. The data can be accessed at https://figshare.com/articles/30630713, and the code can be found at https://figshare.com/articles/30630977.

**Funding:** This work was supported by the Sichuan Provincial Drug Administration (grant number 2024012), the Sichuan Provincial Health Care Committee (grant number Chuan Gan Yan 2021-226), the Chengdu Health Commission (grant number 2022005), and the Committee of Drug-Induced Diseases, Chinese Pharmacological Society (grant number ADR2024MS17). The funders had no role in study design, data collection and analysis, decision to publish, or preparation of the manuscript.

**Competing interests:** The authors have declared that no competing interests exist. This does not alter our adherence to PLOS ONE policies on sharing data and materials.

and extrinsic pathways converge. By effectively inhibiting FXa, fondaparinux reduces thrombin generation and subsequently prevents the conversion of fibrinogen to fibrin, thereby achieving a potent anticoagulant effect [9].

Clinical guidelines have demonstrated that fondaparinux provides favorable efficacy and safety across diverse patient populations, including lactating women [10], patients with malignancy [11], obesity [12], lower extremity venous thrombosis [13], and those receiving prophylaxis against VTE [14] or COVID 19 associated thrombosis [15]. Despite these advantages, its interference with the coagulation process inherently increases bleeding risk, which remains the most frequently reported adverse event [16–18]. Mild manifestations such as epistaxis, gingival bleeding, or subcutaneous petechiae are common, whereas severe cases may involve gastrointestinal or intracranial hemorrhage that can be life threatening.

Post marketing pharmacovigilance has therefore become essential to better define the real world safety profile of fondaparinux beyond controlled clinical trials. The FDA Adverse Event Reporting System (FAERS), established in 2004, is a publicly accessible spontaneous reporting database that serves as a cornerstone of global pharmacovigilance [19]. By collecting adverse event reports from varied healthcare settings, FAERS enables detection of rare, unexpected, or delayed reactions that may not emerge in premarketing studies [20].

In this study, we utilized the FAERS database to systematically characterize adverse drug events associated with fondaparinux using disproportionality analysis. The large, real world dataset allows identification of both expected and potential novel safety signals. Given the inherent limitations of spontaneous reporting systems such as underreporting, missing exposure data, and potential confounding our findings should be interpreted as hypothesis generating, warranting further validation through epidemiologic and mechanistic studies.

## 2. Methods

### 2.1. Source of data

This retrospective pharmacovigilance study was based on FAERS data covering the period from Q1 2004 to Q3 2024. FAERS is maintained by the U.S. Food and Drug Administration and is publicly accessible via the FDA's website [21]. The database includes adverse event reports submitted by a range of sources and is structured for international comparability. Reports were downloaded in ASCII format and processed using R Studio (v4.3). As FAERS contains de identified data, no ethical approval or informed consent was required.

### 2.2. Data processing

Following extraction, duplicate reports were identified based on identical case IDs and removed by retaining only the most recent version by report date. The primary analysis was restricted to records in which fondaparinux was coded as the primary suspect drug (role code "PS") in the DRUG file [22]. Drug names were standardized using the Medex_UIMA_1.8.3 system, and adverse events were encoded using Preferred Terms (PTs) and System Organ Classes (SOCs) from MedDRA version

25.0 [23,24]. Some Preferred Terms (e.g., 'thrombosis with thrombocytopenia syndrome') reflect standardized database terminology rather than distinct clinical entities.

Demographic and clinical features including age, sex, route of administration, geographic region, reporter type, and time to onset were extracted. Disproportionality analysis was then conducted using four established algorithms: Reporting Odds Ratio (ROR), Proportional Reporting Ratio (PRR), Bayesian Confidence Propagation Neural Network (BCPNN), and Empirical Bayesian Geometric Mean (EBGM) [24–29]. The 2×2 contingency framework used to quantify the disproportionality of fondaparinux related adverse events (AEs) compared with non drug related events is detailed in Supplementary Material 1. Signal detection was conducted using complementary algorithms with method specific threshold criteria to ensure broader coverage, cross validation of findings, and reduced false positives, thus enhancing the reliability of the detected safety signals. Thresholds were further adjusted to improve sensitivity for rare adverse events, with the corresponding formulas and criteria provided in Supplementary Material 1. All analyses were conducted using Microsoft Excel (version 2021, Microsoft Corp., Redmond, WA, USA).

## 2.3. Signal filtering and classification

The initial screening included PTs with at least three fondaparinux related AE reports. MedDRA's PTs and SOCs were used to encode, classify, and group the detected signals to identify the major organ systems involved. Similar PTs describing overlapping clinical entities were evaluated collectively to facilitate interpretation.

To ensure methodological transparency and reproducibility, all analytical procedures were implemented using a fully scripted workflow in R Studio. Data extraction, cleaning, and standardization were performed using base R and packages from the tidyverse ecosystem (readr, dplyr, stringr, tidyr).

Duplicate reports were identified by matching CASEID and primaryid, and only the most recent version of each case was retained. Disproportionality analyses were conducted through custom R functions built upon established packages: epiR and DescTools for ROR and PRR computation, bayesAB for BCPNN, and openEBGM (version 0.8.3) for EBGM estimation. Signal detection thresholds followed internationally recognized pharmacovigilance criteria (ROR lower 95% CI > 1, PRR ≥ 2 with $\chi^2 \geq 4$, IC025 > 0, EB05 > 1).

## 2.4. Stratified and exclusion analyses for concomitant therapy

To assess the potential influence of concomitant anticoagulant or antiplatelet therapy, a stratified analysis was performed for key PTs related to bleeding and thrombocytopenia, including hematoma, haemorrhage, muscle haemorrhage, and heparin induced thrombocytopenia (HIT) like reactions.

The proportions of reports that included concomitant exposure to heparin/LMWH, warfarin, direct oral anticoagulants (DOACs), or antiplatelet agents were calculated to estimate the potential contribution of combination therapy.

Subsequently, a separate disproportionality analysis was conducted after excluding all reports containing concomitant anticoagulants or antiplatelet drugs, to evaluate whether the identified signals persisted independently of polypharmacy.

All calculations used the same four algorithms (ROR, PRR, BCPNN, and EBGM) and the same signal detection thresholds as in the main analysis.

## 2.5. Comparative disproportionality analysis across anticoagulant classes

To determine whether the observed bleeding and HIT related signals were drug specific or reflected class effects, a comparative disproportionality analysis was conducted across major anticoagulant classes.

The comparator drugs included enoxaparin, unfractionated heparin (UFH), and direct oral anticoagulants (DOACs: apixaban, dabigatran, edoxaban, and rivaroxaban).

Individual FAERS reports for each agent were extracted using the same inclusion and exclusion criteria and MedDRA mapping procedures applied to fondaparinux. Adverse events related to haemorrhage, haematoma, and HIT were

 

identified based on predefined PTs. Disproportionality was calculated using the same four algorithms (ROR, PRR, BCPNN, EBGM) and uniform signal detection thresholds.

Comparative interpretation focused on identifying whether positive signals occurred consistently across anticoagulant classes (suggesting a class effect) or appeared only with specific agents (indicating agent specific patterns). Rare events such as thrombosis with thrombocytopenia syndrome (TTS) were treated as hypothesis generating due to limited case counts.

## 2.6. Statistical software and tools

All data extraction, processing, statistical analyses, and visualization were performed using R Studio (version 4.3.0) and Microsoft Excel 2021. Custom R scripts were developed for data cleaning, duplicate removal, MedDRA term mapping, and computation of the four disproportionality algorithms (ROR, PRR, BCPNN, and EBGM). Data manipulation and plotting were implemented using the tidyverse ecosystem, including the dplyr, tidyr, and ggplot2 packages. All numerical results were cross checked using Excel for verification and tabulation.

## 2.7. Ethical approval

As FAERS data are de-identified and publicly available, this study did not require institutional ethics approval or informed consent.

## 3. Results

### 3.1. Basic information of the fondaparinux related AEs

A total of 21,483,491 reports were obtained from the FAERS database from the first quarter of 2004 to the third quarter of 2024. As the database is updated quarterly, some reports inevitably overlap with previously published ones, necessitating reprocessing. Following the guidance of the U.S. Food and Drug Administration (FDA), a duplicate data removal process was conducted prior to statistical analysis, reducing the report count to 17,947,757. Among these, 5,788 reports related to fondaparinux with a "Primary Suspected (PS)" role were identified, involving 17,523 AEs. The screening process for fondaparinux related AEs is shown in Fig 1.

The flowchart illustrates the selection procedure of fondaparinux-associated adverse events from the FAERS database and the analytical framework applied for signal detection.

Among the 5,788 adverse event reports related to fondaparinux, 34.97% were male and 48.72% were female, indicating a higher proportion of females. The age distribution of patients showed that the majority were over 65 years old (39.24%). Analysis of the reporting time revealed a sharp increase in fondaparinux related adverse events starting in 2007, peaking at 400 cases in 2009, followed by a sharp decline to 206 cases in 2015, approximately 34% of the peak value. Fig 2 illustrates the quarterly trend of adverse event reports over the years.

The line chart presents the yearly number of adverse event reports in which fondaparinux was identified as the primary suspect drug.

A substantial portion of reports (29.62%) lacked explicit outcome data, hindering a complete evaluation of the clinical consequences associated with fondaparinux use. Among cases where outcomes were reported, the three most frequently documented serious outcomes were hospital admissions (41.84%), medically significant events (29.62%), and fatalities (15.07%). Regarding geographic distribution, the United States contributed the largest number of cases (743), comprising 12.84% of the total. Combined, France, Ireland, and Italy accounted for 17.38% of the reports. However, over half of the cases (66.21%) originated from unspecified regions, which may obscure insights into geographical trends and population level risk patterns.

In terms of reporting identity, consumers were the primary reporters, accounting for 45.91% of the total, while physicians, pharmacists, and other healthcare professionals made up a relatively larger portion, accounting for 51.22%.

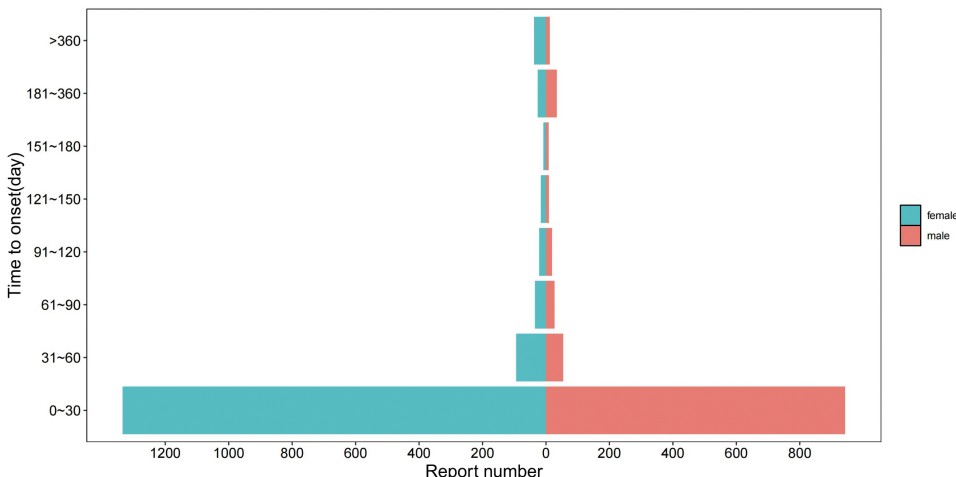

**Fig 1. The flow diagram of selecting fondaparinux-related AEs from FAES database.**

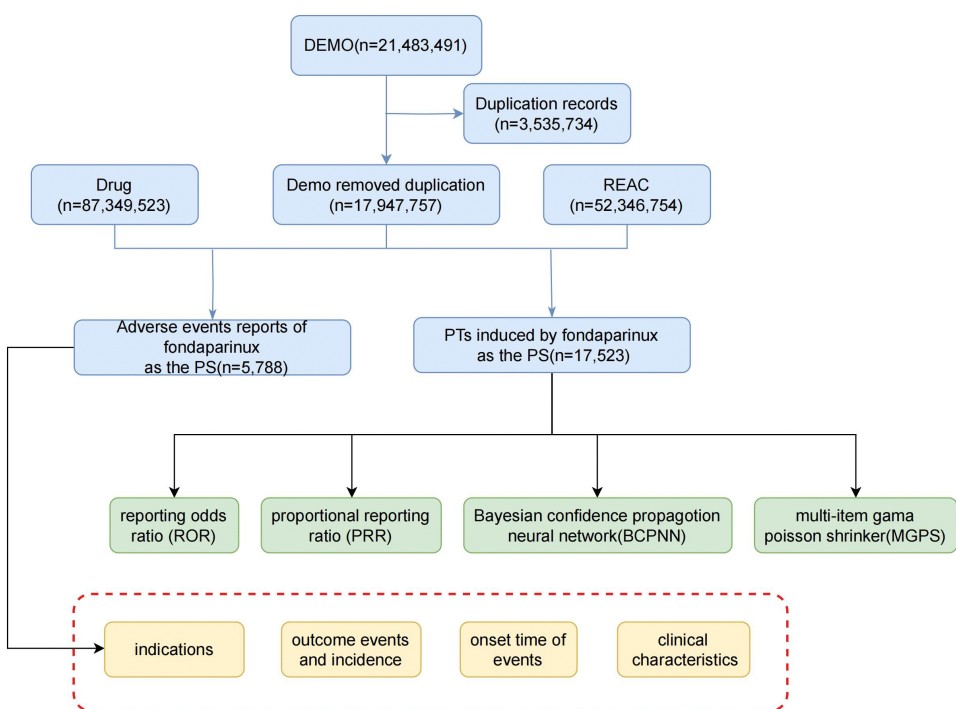

**Fig 2. Distribution of adverse reactions of fondaparinux in each year.**

Notably, the majority of adverse events (72.34%) occurred during subcutaneous injection. Given that fondaparinux has a molecular weight of 1728 Da, it is more readily absorbed through subcutaneous tissue, with a bioavailability close to 100% post subcutaneous injection, making it more convenient and safer compared to intravenous injection [24]. The clinical characteristics of AEs associated with fondaparinux are detailed in Table 1.

**Table 1. Epidemiological characteristics of fondaparinux adverse event reports.**

| variable | Total |
|---|---|
| **Sex** | |
| female | 2820(48.72) |
| male | 2024(34.97) |
| unknown | 944(16.31) |
| **Age (years)** | 69.00(55.00,79.00) |
| **Age group** | |
| <18 | 31(0.54) |
| 18~65 | 1578(27.26) |
| >=65 | 2271(39.24) |
| unknow | 1908(32.96) |
| **Reporter type** | |
| Consumer | 2657(45.91) |
| Physician | 1919(33.15) |
| Pharmacist | 525(9.07) |
| Other health-professional | 521(9.00) |
| unknown | 158(2.73) |
| Lawyer | 4(0.07) |
| Registered Nurse | 4(0.07) |
| **Country of report** | |
| other | 3832(66.21) |
| United States | 743(12.84) |
| France | 504(8.71) |
| Ireland | 305(5.27) |
| Italy | 197(3.40) |
| Germany | 97(1.68) |
| Japan | 59(1.02) |
| United Kingdom | 51(0.88) |
| **Route of administration** | |
| subcutaneous | 4187(72.34) |
| other | 1360(23.50) |
| intravenous | 184(3.18) |
| oral | 22(0.38) |
| transplacental | 20(0.35) |
| intramuscular | 15(0.26) |
| **Outcomes** | |
| hospitalization | 2829(41.84) |
| other serious | 2003(29.62) |
| death | 1019(15.07) |
| life threatening | 654(9.67) |
| disability | 201(2.97) |
| required intervention to Prevent Permanent Impairment/Damage | 36(0.53) |
| congenital anomaly | 20(0.30) |
| **Time to onset (days)** | 5.00(1.00,14.00) |
| 0~30 | 2497(57.48) |
| 31~60 | 155(3.57) |

*(Continued)*

**Table 1.** (Continued)

| variable | Total |
|---|---|
| 61~90 | 64(1.47) |
| 91~120 | 41(0.94) |
| 121~150 | 26(0.60) |
| 151~180 | 17(0.39) |
| 181~360 | 66(1.52) |
| >360 | 52(1.20) |
| unknow | 1426(32.83) |

Values are presented as n (%) unless otherwise indicated. Continuous variables such as Age (years) and Time to onset (days) are expressed as median (interquartile range). Time to onset category represents the grouped distribution of onset intervals.

### 3.2. Signal detection of fondaparinux related AEs

**3.2.1. Analysis by SOC level.** All reported adverse events were classified according to 25 distinct SOCs. A complete breakdown of the signal strengths across all SOC categories is provided in Table 2. Among these, only two SOCs demonstrated consistently elevated metrics across all four detection algorithms. Specifically, "vascular disorders" exhibited strong associations with fondaparinux, supported by 2,046 cases and corresponding values of ROR 5.82, PRR 5.26, IC 2.39, and EBGM 5.25. Likewise, "blood and lymphatic system disorders" emerged as another key domain, with 1,125 reported cases and signal indices of ROR 3.85, PRR 3.66, IC 1.87, and EBGM 3.66. A detailed comparison is shown in Table 2.

**3.2.2. Analysis by PT level.** After applying the predefined thresholds of all four disproportionality algorithms, a total of 259 PTs were found to be positively associated with fondaparinux use. These signals were distributed across 23 different SOCs. The complete list of PT level signals that met the criteria is presented in Supplementary Material 2. Among the individual PTs based on ROR values, two signals related to fondaparinux use were particularly prominent: abnormal coagulation factor X concentration (ROR 1792.1) and thrombosis with TTS (ROR 543.09). Clinically, this PT describes reports presenting concurrent thrombosis and thrombocytopenia, which may correspond to conditions such as HIT, vaccine induced thrombotic thrombocytopenia (VITT), or other PF4 related disorders. It is noteworthy that certain signals were based on a limited number of reports some appearing in just three or four cases making their clinical implications uncertain. Despite this, hematoma (582 cases), anemia (523), bleeding (341), pulmonary embolism (296), decreased hemoglobin (291), and muscle hematoma (206) were among the more frequently reported adverse events. Although the statistical signal strength for muscle hematoma was relatively lower compared to the others, the volume of reported cases still suggests that this event deserves clinical attention. Fig 3 visualizes the most frequently reported Preferred Terms within the most affected System Organ Classes.

Panels A–F display the main adverse events detected for fondaparinux across different organ systems, including injury and procedural complications, gastrointestinal disorders, nervous system disorders, investigations, blood and lymphatic system disorders, and vascular disorders. The horizontal axis represents the measure of disproportionality, the vertical axis indicates statistical strength, and the bubble size reflects the number of reported cases.

**3.2.3. Stratified and comparative analysis across anticoagulant exposures.** A stratified analysis was conducted to evaluate the potential influence of concomitant anticoagulant and antiplatelet exposure on major fondaparinux associated adverse events, including hematoma, haemorrhage, muscle haemorrhage, and thrombosis with thrombocytopenia events. Among hematoma reports (n = 582), 2.6% included heparin or low molecular weight heparin (LMWH), 3.3% warfarin, 0.9% DOACs, and 10.5% antiplatelet agents. For HIT (n = 115), 26.1% involved heparin/LMWH and 6.1% warfarin. In contrast,

**Table 2. The signal strength of AEs of fondaparinux at the SOC level in FAERS database.**

| SOC | Case Reports | ROR(95% CI) | PRR(95% CI) | chisq | IC(IC025) | EBGM(EBGM05) |
|---|---|---|---|---|---|---|
| vascular disorders | 2046 | 5.82(5.56, 6.09) | 5.26(5.06, 5.47) | 7197.64 | 2.39(2.33) | 5.25(5.05) |
| blood and lymphatic system disorders | 1125 | 3.85(3.62, 4.09) | 3.66(3.45, 3.88) | 2214.47 | 1.87(1.78) | 3.66(3.48) |
| investigations | 1627 | 1.5(1.43, 1.58) | 1.45(1.39, 1.51) | 247.14 | 0.54(0.47) | 1.45(1.39) |
| renal and urinary disorders | 412 | 1.25(1.13, 1.38) | 1.24(1.12, 1.37) | 19.67 | 0.31(0.17) | 1.24(1.14) |
| hepatobiliary disorders | 194 | 1.18(1.02, 1.36) | 1.18(1.03, 1.35) | 5.12 | 0.23(0.03) | 1.18(1.04) |
| pregnancy, puerperium and perinatal conditions | 91 | 1.17(0.95, 1.44) | 1.17(0.96, 1.42) | 2.28 | 0.23(−0.07) | 1.17(0.99) |
| gastrointestinal disorders | 1769 | 1.16(1.11, 1.22) | 1.15(1.11, 1.2) | 36.54 | 0.2(0.13) | 1.15(1.1) |
| injury, poisoning and procedural complications | 1863 | 1.12(1.07, 1.18) | 1.11(1.07, 1.15) | 21.64 | 0.15(0.08) | 1.11(1.06) |
| respiratory, thoracic and mediastinal disorders | 911 | 1.06(0.99, 1.14) | 1.06(1, 1.12) | 3.12 | 0.08(−0.01) | 1.06(1) |
| cardiac disorders | 500 | 1.04(0.95, 1.14) | 1.04(0.96, 1.12) | 0.77 | 0.06(−0.07) | 1.04(0.96) |
| nervous system disorders | 1582 | 1.04(0.98, 1.09) | 1.03(0.99, 1.07) | 1.89 | 0.05(−0.03) | 1.03(0.99) |
| surgical and medical procedures | 204 | 0.84(0.73, 0.96) | 0.84(0.73, 0.96) | 6.51 | −0.26(−0.45) | 0.84(0.75) |
| skin and subcutaneous tissue disorders | 774 | 0.79(0.74, 0.85) | 0.8(0.74, 0.87) | 41.23 | −0.32(−0.43) | 0.8(0.75) |
| musculoskeletal and connective tissue disorders | 734 | 0.76(0.71, 0.82) | 0.77(0.71, 0.83) | 53.16 | −0.38(−0.48) | 0.77(0.72) |
| general disorders and administration site conditions | 2292 | 0.69(0.66, 0.72) | 0.73(0.7, 0.76) | 277.78 | −0.45(−0.52) | 0.73(0.7) |
| congenital, familial and genetic disorders | 32 | 0.57(0.41, 0.81) | 0.57(0.4, 0.81) | 10.14 | −0.8(−1.29) | 0.57(0.43) |
| endocrine disorders | 25 | 0.55(0.37, 0.81) | 0.55(0.37, 0.81) | 9.45 | −0.87(−1.43) | 0.55(0.39) |
| reproductive system and breast disorders | 76 | 0.51(0.41, 0.64) | 0.51(0.41, 0.63) | 35.81 | −0.97(−1.29) | 0.51(0.42) |
| neoplasms benign, malignant and unspecified (incl cysts and polyps) | 243 | 0.5(0.44, 0.57) | 0.51(0.45, 0.57) | 119.49 | −0.98(−1.16) | 0.51(0.46) |
| metabolism and nutrition disorders | 175 | 0.45(0.38, 0.52) | 0.45(0.38, 0.53) | 119.1 | −1.15(−1.36) | 0.45(0.4) |
| immune system disorders | 77 | 0.38(0.31, 0.48) | 0.38(0.31, 0.47) | 76.63 | −1.38(−1.7) | 0.38(0.32) |
| eye disorders | 139 | 0.38(0.32, 0.45) | 0.38(0.32, 0.44) | 140.07 | −1.38(−1.62) | 0.38(0.33) |
| infections and infestations | 359 | 0.37(0.33, 0.41) | 0.38(0.34, 0.42) | 388.07 | −1.4(−1.55) | 0.38(0.35) |
| ear and labyrinth disorders | 28 | 0.36(0.25, 0.52) | 0.36(0.25, 0.52) | 32.01 | −1.47(−2) | 0.36(0.26) |
| psychiatric disorders | 245 | 0.23(0.2, 0.26) | 0.24(0.21, 0.27) | 635 | −2.07(−2.25) | 0.24(0.21) |

SOC = System Organ Class; ROR = reporting odds ratio; PRR = proportional reporting ratio; IC = information component (BCPNN method), with IC025 indicating its lower 95% credibility bound; EBGM = empirical Bayesian geometric mean, with EBGM05 indicating its lower confidence bound. Disproportionality metrics represent the relative reporting strength of AEs at the SOC level in the FAERS database.

most reports of muscle haemorrhage (81.6%) and haemorrhage (84.5%) did not involve concomitant anticoagulant or antiplatelet therapy, suggesting that these signals were not solely attributable to polypharmacy. The detailed distribution of concomitant drug exposures is presented in Table 3.

To minimize confounding from background bleeding risks, a separate disproportionality analysis was performed after excluding all reports containing other anticoagulants or antiplatelet agents. Significant signals persisted across all major bleeding related PTs hematoma (ROR = 66.78 [60.99–73.12]), muscle haemorrhage (ROR = 161.13 [137.88–188.30]), and haemorrhage (ROR = 10.04 [8.94–11.28]) indicating that these associations were largely independent of combined therapy. Consistent results were confirmed across PRR, IC, and EBGM algorithms. For HIT (ROR = 52.00 [41.37–65.36]) and thrombosis with TTS (n = 4), the limited number of cases precluded robust estimation. Full results for the sensitivity analysis excluding concomitant anticoagulants and antiplatelets are summarized in Table 4.

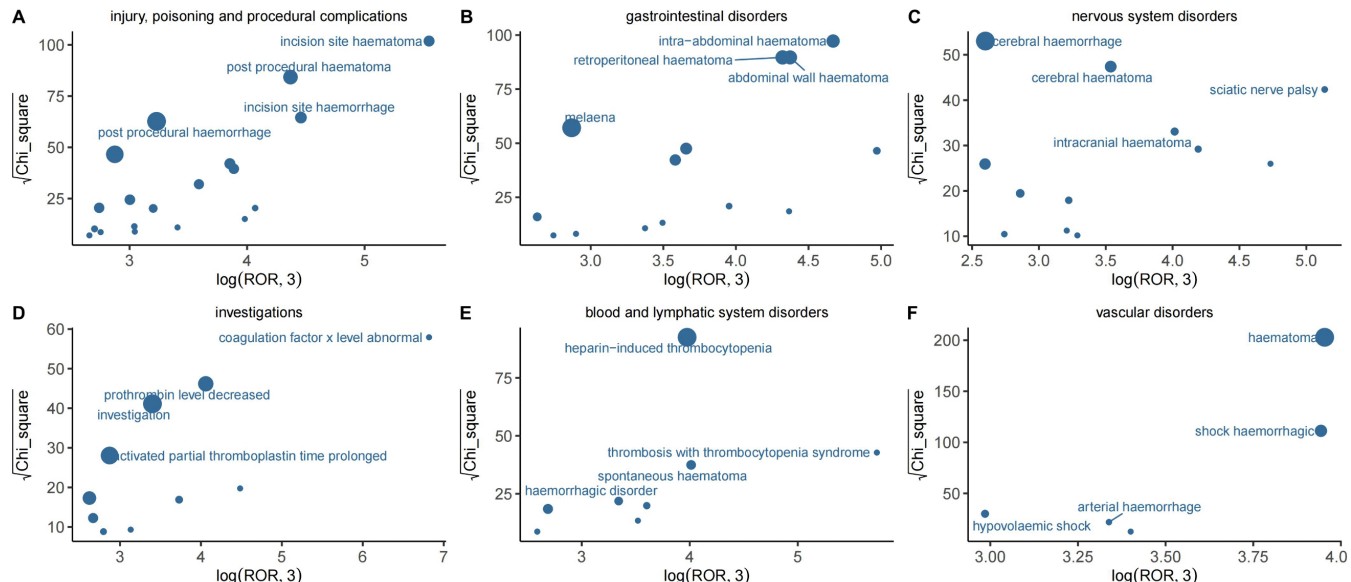

**Fig 3. The most relevant preferred terms within the most significant System Organ Classification in fondaparinux.**

**Table 3. Stratified analysis of concomitant anticoagulant and antiplatelet exposure in fondaparinux-related major adverse events.**

| | Hematoma (N=582) | | HIT (N=115) | | TTS (N=4) | | Muscle Haemorrhage (N=206) | | Haemorrhage (N=341) | |
|---|---|---|---|---|---|---|---|---|---|---|
| **Concomitant medication** | Count | Percent | Count | Percent | Count | Percent | Count | Percent | Count | Percent |
| Heparin/LMWH | 15 | 2.6 | 30 | 26.1 | 1 | 25 | 3 | 1.5 | 10 | 2.9 |
| Warfarin | 19 | 3.3 | 7 | 6.1 | 2 | 50 | 4 | 1.9 | 18 | 5.3 |
| DOACs | 5 | 0.9 | 1 | 0.9 | 1 | 25 | 2 | 1 | 2 | 0.6 |
| Other anticoagulants | 0 | 0 | 5 | 4.3 | 3 | 75 | 0 | 0 | 1 | 0.3 |
| Antiplatelets | 61 | 10.5 | 6 | 5.2 | 0 | 0 | 32 | 15.5 | 28 | 8.2 |
| No concomitant anticoagulant/antiplatelet | 490 | 84.2 | 75 | 65.2 | 0 | 0 | 168 | 81.6 | 288 | 84.5 |

Concomitant medications were grouped as heparin/LMWH, warfarin, DOACs, other anticoagulants, and antiplatelet agents. Percentages indicate the proportion within each adverse event category. HIT = heparin induced thrombocytopenia; TTS = thrombosis with thrombocytopenia syndrome.

**Table 4. Disproportionality analysis of major fondaparinux-associated adverse events after excluding concomitant anticoagulant and anti-platelet therapies.**

| | Hematoma (N=582) | Hit (N=115) | Tts (N=4) | Muscle Haemorrhage (N=206) | Haemorrhage (N=341) |
|---|---|---|---|---|---|
| ROR(95% CI) | 66.78(60.99,73.12) | 52(41.37,65.36) | – | 161.13(137.88,188.3) | 10.04(8.94,11.28) |
| PRR(95% CI) | 64.95(59.46,70.94) | 51.78(41.23,65.03) | – | 159.61(136.77,186.26) | 9.89(8.82,11.1) |
| chisq | 30226.3 | 3673.18 | – | 25168.36 | 2298.7 |
| IC(IC025) | 5.99(4.33) | 5.67(4) | – | 7.25(5.58) | 3.3(1.64) |
| EBGM(EBGM05) | 63.62(58.97) | 50.94(42.06) | – | 151.75(133.2) | 9.86(8.95) |

Analyses exclude reports involving any anticoagulant or antiplatelet drugs. ROR = reporting odds ratio; PRR = proportional reporting ratio; IC = information component; EBGM = empirical Bayesian geometric mean. "–" indicates insufficient data for computation.

To further characterize whether these signals represented drug specific or class wide effects, a comparative disproportionality analysis was performed across other anticoagulant classes, including enoxaparin, UFH, and DOACs (apixaban, dabigatran, edoxaban, and rivaroxaban). All agents demonstrated strong positive signals for haemorrhage and haematoma, with RORs generally ranging from 8 to 17 for DOACs, 17–70 for enoxaparin, and 8–13 for UFH, confirming a class effect associated with anticoagulant therapy. In contrast, disproportionate reporting of HIT related events was observed exclusively for fondaparinux (ROR ≈ 52) and UFH (ROR ≈ 1773), whereas DOACs showed no significant disproportionality (ROR ≈ 1). These findings reinforce that bleeding signals likely reflect a class wide anticoagulant effect, while heparin induced thrombocytopenia remains a heparin specific phenomenon. Detailed comparative results for each anticoagulant agent are provided in Supplementary Material 3.

**3.2.4. Grouped by age.** To investigate the age related safety profile of fondaparinux, we performed a stratified analysis based on age categories. The findings across different age groups are summarized in Supplementary Material 4. Reports involving individuals under the age of 18 were excluded from detailed discussion due to the minimal number of cases, which limits interpretability and clinical relevance. This decision aligns with the drug's prescribing information, which states that safety data are lacking for patients below 17 years of age.

Among adult patients, a total of 101 and 100 Preferred Terms were identified in the 18–65 and >65 year age groups, respectively. Hematoma was the most frequently reported adverse event across both age segments (413 cases), indicating its potential as a core safety concern in the adult population. Notably, in the 18–65 group, although TTS was reported in only four instances, it demonstrated the highest disproportionality signal (ROR 1102.84), far surpassing other PTs. Hematoma, however, remained the most common event (93 cases) in this group. In patients older than 65, hematoma also ranked highest in frequency (320 reports), while incision site hematoma presented the strongest signal (ROR 201.01). These results suggest that hematoma, regardless of age, warrants vigilant clinical monitoring in individuals receiving fondaparinux.

**3.2.5. Grouped by gender.** Fig 4 illustrates the gender differences between the two treatment regimens, with detailed information provided in Supplementary Material 5 In the fondaparinux treatment group, Hematoma was identified as the most prominent adverse event across both sexes based on combined evaluation of signal strength and report counts. Notably, female patients reported this event more frequently (325 cases) than their male counterparts (166 cases). Additionally, muscle hematoma (120 cases) was more commonly reported in females, while anemia was more frequently observed in males (174 cases).

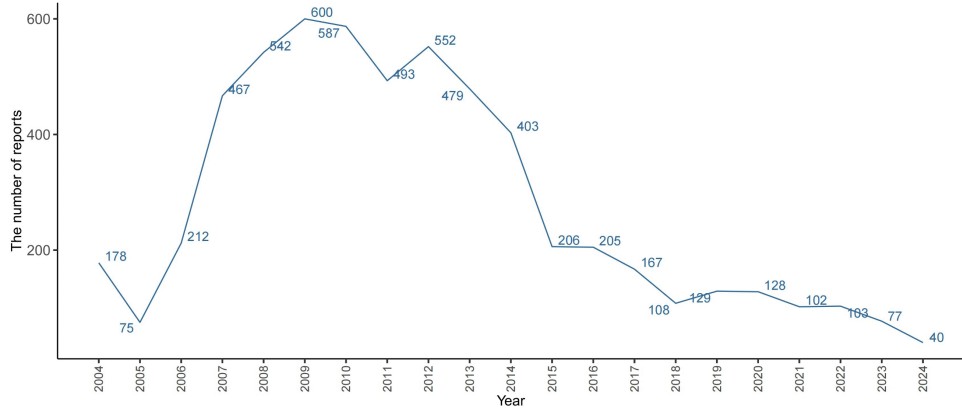

**Fig 4. The primary adverse effects of fondaparinux treatment in men and women.**

The plot compares the strength and frequency of reported adverse events between men and women receiving fondaparinux. Each point represents an individual event, with its position indicating the relative difference between the two sexes.

**3.2.6. Time to onset analysis.** Among all AE reports, after excluding those with inaccurate or missing onset times, a total of 2,917 reports included information on the timing of AE onset. Fig 5 illustrates that most adverse events in both male and female patients occurred within the first month of initiating treatment. This pattern is consistent with the standard clinical duration of fondaparinux therapy. Although a small number of events were reported beyond the initial treatment period, these may reflect delayed reporting rather than delayed drug effects. Therefore, it remains important to maintain vigilance for adverse events during the active treatment phase and shortly after drug discontinuation.

The figure presents the number of adverse event reports grouped by the interval between treatment initiation and event onset. Most cases in both men and women occurred within the first 30 days after starting treatment.

## 4. Discussion

The FAERS database is a spontaneous reporting system that allows the detection of disproportional reporting signals but does not establish causal relationships between drug exposure and adverse events. Therefore, all findings in this study should be interpreted as associations rather than confirmed causality.

Previous studies on fondaparinux related AEs have primarily relied on clinical trials and case reports. However, clinical studies often have stringent trial designs and inclusion criteria, which may lead to underreporting of AEs, resulting in incomplete drug safety information. This study utilized a large real world dataset, analyzing fondaparinux related AEs over the past 20 years using the FAERS database. Additionally, we investigated the differences between AEs reported in the approved product label and those observed in real world settings, identifying rare and potential AEs.

To further distinguish between known and potentially novel adverse events, all significant PT level signals identified in this study were cross checked against the current FDA prescribing information for fondaparinux (label revision 2024). The key events such as hemorrhage, hematoma, and thrombocytopenia are consistent with labeled adverse reactions,

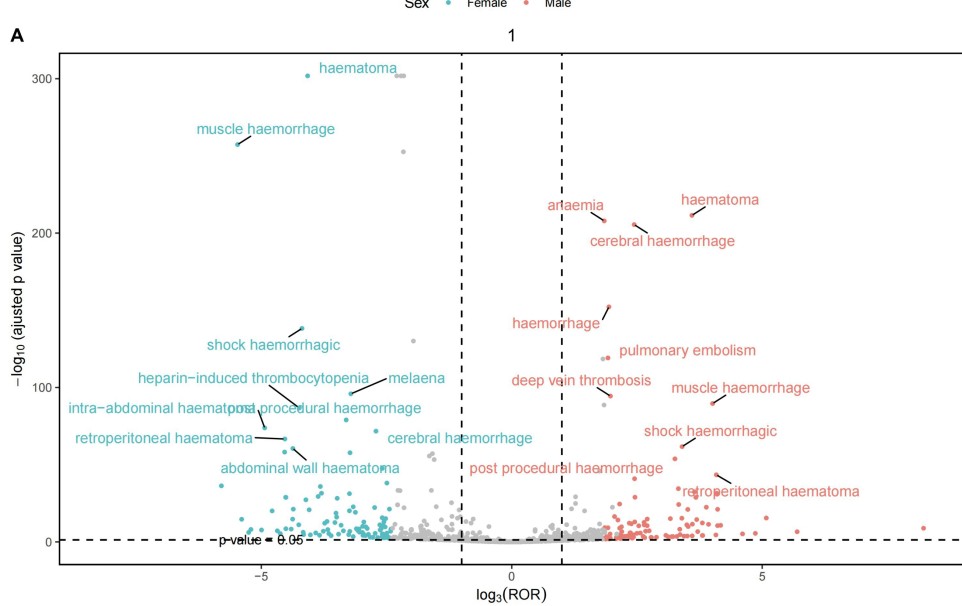

**Fig 5. Time to Onset of Adverse Reactions Associated with Fondaparinux.**

confirming that FAERS based analyses can reliably capture known safety signals. In contrast, several rare PTs including abnormal coagulation factor X concentration and thrombosis associated with thrombocytopenia are not explicitly described in the FDA label, suggesting they may represent underrecognized or emerging adverse events. However, given the small number of reported cases and the inherent limitations of spontaneous reporting systems, these findings should be interpreted cautiously as hypothesis generating observations rather than confirmed risks [30]. Between 2007 and 2014, the volume of AE reports for fondaparinux peaked. This is likely due to the approval of fondaparinux by the European Medicines Agency (EMEA) in 2007 for the new indication of acute coronary syndrome (ACS) [31]. As clinical experience with fondaparinux continues to accumulate, its associated AEs have gained increasing attention, and corresponding preventive measures are gradually being improved. Existing studies have shown that, compared to low molecular weight heparin, fondaparinux offers greater efficacy in reducing the risk of perioperative venous thromboembolism. However, it is also associated with an increased risk of major bleeding [32]. The marked decline in fondaparinux related AE reporting after 2015 likely reflects two concurrent trends. First, clinical utilization of fondaparinux has progressively decreased as DOACs became the preferred anticoagulants in many thromboembolic indications. Second, spontaneous reporting typically diminishes once a drug becomes well established and its safety characteristics are familiar to prescribers, reducing the motivation to report. These factors together may explain the sustained downward trend observed in recent years.

In this study, the number of fondaparinux related AEs was similar in both males (2024 cases) and females (2820 cases). However, in the "vascular disorders" category, females had a higher number of reports for certain bleeding related AEs, such as hematoma, with 325 cases in females compared to only 166 in males. This difference may be related to factors such as the female physiological cycle and hormone levels [33]. Older patients aged 65 and above experienced a higher number of fondaparinux related AEs. In the elderly population, changes in the body's hemostatic balance occur, with enhanced coagulation function and reduced fibrinolytic activity, leading to an increased risk of thrombosis. At the same time, the risk of bleeding is also higher during anticoagulant therapy [34,35]. Additionally, conditions such as blood stasis, vascular wall degeneration, and endothelial dysfunction are more prevalent in the elderly. These conditions lead to increased platelet activation, which in turn promotes arterial thrombosis, thereby raising the risk of bleeding [36,37]. Furthermore, elderly individuals often experience renal impairment, and since fondaparinux is mainly eliminated through renal excretion. Impaired kidney function may reduce clearance, thereby elevating the risk of bleeding.In Europe, for elderly patients with moderate renal impairment (creatinine clearance of 20–50 mL/min), a reduced dose of 1.5 mg/day has been approved for the prevention of venous thromboembolism (VTE) [31]. Studies have shown that this dosage provides good efficacy and safety; however, its effectiveness and safety in patients with severe renal impairment still require further validation. In the United States, fondaparinux is contraindicated in patients with severe renal impairment (creatinine clearance <30 mL/min) for VTE prevention and treatment [30,38].

The majority of AE reporters are consumers and physicians. This suggests that during clinical use, fondaparinux is closely monitored by healthcare professionals. Meanwhile, consumers, as direct users of the medication, are also proactive in reporting adverse reactions to ensure their own medication safety. Apart from reports from unknown sources, the United States has the highest number of submissions. The higher number of reports from the United States is primarily due to the use of FAERS, which is the spontaneous reporting system (SRS) database for the U.S. Only serious and/or unknown adverse events reported worldwide and collected in the WHO VigiBase are transferred to FAERS. In 2017, the global sales of fondaparinux injection reached approximately $190 million, with sales in the US market amounting to $69.53 million (IQVIA, 2018). This has resulted in a longer period of use and a broader clinical application base in the United States, where both physicians and patients have a relatively higher level of awareness and acceptance. In thrombosis prevention following major orthopedic surgeries, patients undergoing total hip or knee arthroplasty typically start fondaparinux treatment 6–8 hours post surgery, continuing for 5–10 days, while hip fracture surgery patients follow a 2–4 week regimen. Adverse reactions related to fondaparinux can still emerge during the first month of therapy. This underscores the importance of ongoing monitoring even after the initial phase of treatment.

Our findings confirm that the two most relevant SOCs associated with fondaparinux are vascular disorders and blood and lymphatic system disorders, which is largely consistent with what is mentioned in the drug's prescribing information.

### 4.1. vascular disorders

In terms of correlation, hemorrhagic shock showed a ROR of 76.09 (65.38, 88.54) and a PRR of 75.34 (64.41, 88.13), both with relatively high values and narrow confidence intervals. This indicates a strong association between the use of fondaparinux and hemorrhagic shock. Such events are pharmacologically expected for anticoagulant agents and represent well established adverse reactions rather than novel safety signals. Despite clinical evidence indicating that fondaparinux carries a lower incidence of major bleeding compared to enoxaparin (2.1% versus 4.1%), fatal bleeding events such as hemorrhagic shock have still been reported, albeit infrequently (0.3%) [39]. Fondaparinux does not have a targeted anti-dote, and conventional agents such as vitamin K and protamine sulfate have proven ineffective in mitigating its anticoagulant activity [40]. Findings from in vitro coagulation studies indicate that partial reversal may be achieved through the administration of activated prothrombin complex concentrate (aPCC) or recombinant activated factor VII (rFVIIa), while dialysis appears to provide limited removal of the drug from circulation [41,42]. Among healthy individuals receiving standard therapeutic doses, a high concentration of rFVIIa (90 µg/kg) was shown to partially restore coagulation parameters, including activated partial thromboplastin time (aPTT), endogenous thrombin potential, and markers of prothrombin activation [40]. Furthermore, a clinical case documented the onset of hemorrhagic shock following a 2.5 mg dose of fondaparinux, where hemostatic control was successfully achieved through the combined use of rFVIIa and tranexamic acid. [43].

In terms of the number of case reports, hematoma had the highest number of reports, reaching 582 cases. Fondaparinux reduces thrombin generation by inhibiting factor Xa, disrupting the coagulation cascade, and prolonging clotting time. Even at therapeutic doses, excessive suppression of coagulation can lead to spontaneous bleeding or bleeding after minor trauma, resulting in subcutaneous, muscular, or deep tissue hematomas. In a study of Asian populations for VTE prevention following total knee and total hip arthroplasty, fondaparinux showed a higher relative risk (RR) for minor bleeding: 2.71 (1.12, 6.56) [44], though its incidence varied across different surgeries and studies [45,46]. Hemorrhage had 341 case reports, also a relatively high number. Studies have shown that among adverse event reporting rates for heparins and their derivatives, hemorrhage has the highest incidence, ranging from 2.8 to 140.1/100,000 standard units (SU), with fondaparinux having a relatively higher reporting rate [47]. Similar to hematoma, the higher number of AE case reports provides real world evidence for further investigation into the mechanisms, risk factors, and effective hemostatic measures for bleeding.

### 4.2. blood and lymphatic system disorders

In terms of correlation, the ROR for TTS was 543.09 (187.13, 1576.17) and the PRR was 542.97 (188.42, 1564.71), both of which were extremely high, with relatively wide confidence intervals. This suggests a strong but uncertain association between this adverse event and the use of fondaparinux. Given the very limited number of reports, this finding should be regarded as an exploratory, hypothesis generating signal rather than a confirmed association. For patients suspected of developing thrombosis with thrombocytopenic syndrome, clinicians must thoroughly document the patient's complete medication history, particularly the use of heparins and vaccines, as this condition shares clinical symptoms with HIT and VITT [48,49]. This will allow for the timely identification of the adverse reaction triggers and prevent recurrence.

In terms of case report numbers, anaemia was the most frequently reported adverse event, with 523 cases. A study investigating the effects of anticoagulants on anemia after total hip arthroplasty (THA) [50], reported that 24.4% of patients in the fondaparinux group (comprising 86 individuals) developed postoperative anemia, a finding that aligns with the results observed in the present study. Thrombocytopenia was reported in 150 cases. Platelets play a crucial role in coagulation, and the potent inhibition of factor Xa by fondaparinux alters the coagulation cascade, potentially indirectly affecting platelet activation, aggregation, and adhesion, which may lead to a reduction in platelet count [51]. These events

are pharmacologically expected consequences of anticoagulant therapy and represent well established adverse reactions rather than new safety concerns. HIT was reported in 115 cases. HIT is an immune mediated response initiated by the formation of a complex between heparin and platelet factor 4 (PF4). This interaction results in platelet activation, aggregation, and depletion, accompanied by a heightened risk of thrombosis, distinguishing it from uncomplicated thrombocytopenia. Fondaparinux typically does not induce the classic HIT syndrome because its molecular structure lacks the key site that can specifically bind to PF4. However, although extremely rare, some case reports indicate that fondaparinux may induce HIT [52–54]. Further investigations have found that the potential mechanism shares some similarities with delayed HIT. In these special cases, even after discontinuing heparin, antibodies triggered by prior heparin exposure or other unknown factors may still cause thrombocytopenia and thrombosis. Specifically, when antibodies produced by the body recognize PF4 bound to endogenous platelet related chondroitin sulfate, it triggers a complex immune cascade, leading to the release of more procoagulant substances and inflammatory mediators from platelets, further promoting platelet aggregation. Once HIT occurs, not only is platelet count reduced, but platelet activation may also occur, increasing the risk of thrombosis and causing severe thrombotic complications. Although this phenomenon appears to be extremely rare, it highlights a potential immunologic mechanism that warrants further investigation in future mechanistic or clinical studies.

To better differentiate drug specific effects from the confounding influence of co administered antithrombotics, a restricted disproportionality analysis was performed after excluding reports containing other anticoagulants (UFH, LMWH, warfarin, or DOACs) or antiplatelet agents. The signals for Haemorrhage and thrombocytopenia persisted with statistical significance, indicating that these associations are not merely consequences of combined therapy but may reflect an intrinsic hematologic impact of fondaparinux monotherapy. This finding aligns with post marketing pharmacovigilance studies showing that hematologic signals of fondaparinux remain detectable even after adjustment for concomitant heparin or warfarin exposure [55]. In contrast, signals such as HIT and TTS became less stable, suggesting that such rare immune mediated events are likely confounded by prior heparin sensitization or perioperative conditions rather than direct drug effect.

We also performed a stratified comparison of reports involving concomitant antiplatelet therapy (e.g., aspirin, clopidogrel, ticagrelor). In these reports, hemorrhagic and thrombocytopenia related signals appeared stronger than in fondaparinux monotherapy, suggesting a potential additive pharmacodynamic interaction affecting platelet function or hemostasis. This pattern may stem from overlapping mechanisms of platelet inhibition and anticoagulation [56]. Consistent observations have been described in clinical registries and pharmacovigilance analyses of combined antithrombotic therapy, indicating an overall trend toward increased bleeding and cytopenic risks relative to single agent use [57]. These findings reinforce the need for cautious use of fondaparinux in combination antithrombotic regimens, particularly in elderly or renally impaired patients.

From a regulatory perspective, both the U.S. FDA and the European Medicines Agency (EMA) include haemorrhage and thrombocytopenia in the safety labeling of fondaparinux and emphasize renal function monitoring and individualized risk assessment [30,58]. These requirements reflect the ongoing regulatory attention to hematologic safety profiles of factor Xa inhibitors in general, particularly in patients receiving concomitant antithrombotic therapy or with impaired renal function. The persistence of these hematologic signals in our monotherapy restricted analyses supports current recommendations for laboratory monitoring (e.g., hemoglobin, platelet count) during early treatment and provides complementary postmarketing evidence for the importance of individualized risk evaluation in clinical practice.

### 4.3. Rare reports

For clinicians, focusing solely on the significance at the SOC level may overlook specific but potentially serious adverse events. In this study, although only three reports described abnormal concentrations of coagulation factor X, an extremely high ROR (1792.1) was observed. This disproportionate signal is likely pharmacologically driven, as fondaparinux exerts its anticoagulant effect by selectively inhibiting factor Xa activity [59]. Likewise, thrombosis associated with

thrombocytopenia was reported in four cases, with a markedly elevated ROR (543.09), possibly indicating an immune mediated mechanism that disrupts platelet and coagulation balance. However, because of the very limited number of cases and the wide confidence intervals, these findings should be interpreted as exploratory and hypothesis generating, pending confirmation in larger datasets or clinical investigations [60].

Through a comprehensive analysis of PT data, several adverse events not mentioned in the drug's label were also identified, including eosinophilia (21 cases) [ROR 4.12 (2.69, 6.33), PRR 4.12 (2.68, 6.34)]. Eosinophilia may be associated with allergic reactions, parasitic infections, certain autoimmune diseases, and other conditions. Within the scope of blood and lymphatic system disorders, these findings may indicate that the medication triggered an immune reaction, resulting in an abnormal elevation of eosinophil levels [61,62]. Thrombocytosis was reported in 20 cases [ROR 19.15 (12.33, 29.72), PRR 19.12 (12.42, 29.43)]. Thrombocytosis generally increases the risk of thrombosis, and its occurrence may be related to the drug's impact on bone marrow hematopoiesis or immune feedback regulation mechanisms [63]. The study also identified leukocytosis in 19 cases [ROR 3.55 (2.26, 5.57), PRR 3.55 (2.26, 5.57)], normochromic normocytic anemia in 13 cases [ROR 14 (8.12, 24.15), PRR 13.99 (8.08, 24.22)], and antiphospholipid syndrome in 6 cases [ROR 12.99 (5.82, 28.96), PRR 12.98 (5.81, 28.99)]. Although none of these associations have been documented in prior clinical studies, they represent weak, hypothesis generating signals that warrant pharmacovigilance attention and further mechanistic validation.

### 4.4. Clinical interpretation of time to onset patterns

The time to onset (TTO) analysis revealed that most fondaparinux related adverse events occurred within the first month after treatment initiation. This early onset clustering reflects that adverse events tend to emerge soon after therapy begins, although the pharmacokinetic profile of fondaparinux indicates an elimination half life of approximately 18 hours and near complete clearance within several days. Therefore, the observed time distribution likely represents the clinical treatment window during which the drug is actively administered, rather than a direct pharmacokinetic effect. Bleeding and hematoma events were mainly concentrated in this early phase, particularly among elderly patients or those with renal impairment, where delayed clearance and cumulative exposure may occur. These findings highlight the importance of close clinical and laboratory monitoring during the early treatment period, especially when fondaparinux is used concomitantly with other antithrombotic agents.

In contrast, a small number of delayed onset cases (> 30 days) were identified, which may be explained by prolonged reporting intervals, ongoing follow up after hospital discharge, or rare immune mediated mechanisms such as heparin independent thrombocytopenia. Although such events are uncommon, they highlight the necessity for continued vigilance and careful assessment of hematologic parameters even after discontinuation in selected high risk patients.

### 4.5. Limitations

This study has several limitations that should be acknowledged.

First, the FAERS database is a spontaneous reporting system that is inherently subject to underreporting, duplication, reporting bias, and missing clinical information. Because detailed patient level data (e.g., laboratory results, comorbidities, concomitant medications, and dosages) are unavailable, it is difficult to control for potential confounders or to establish causal relationships between fondaparinux and reported adverse events [64]. Therefore, the present findings can only indicate statistical associations rather than causality.

Second, the FAERS database does not provide drug exposure data (i.e., the number of patients who received fondaparinux); thus, the true incidence or relative risk of adverse events cannot be estimated. The disproportionality metrics applied in this study (ROR, PRR, BCPNN, and EBGM) are suitable for detecting disproportionate reporting patterns but should not be interpreted as direct measures of risk.

Third, the FAERS system lacks information on patients' clinical backgrounds and treatment contexts, making it impossible to distinguish drug related events from those associated with underlying conditions or procedural risks. For instance, fondaparinux is frequently administered to surgical patients for thromboprophylaxis, among whom hematoma or hemorrhage is relatively common. Therefore, postoperative bleeding events cannot be clearly separated from drug specific adverse reactions in this dataset.

Fourth, several safety signals identified in this analysis were based on a small number of reports, limiting statistical power and the robustness of these findings. Such rare or exploratory signals should be interpreted cautiously and validated through prospective studies or analyses using electronic health records.

Fifth, potential reporting bias and confounding should be acknowledged. A large proportion of reports were submitted by consumers rather than healthcare professionals, which may affect data accuracy and clinical interpretation.

Sixth, approximately 30% of the included reports lacked outcome data, which may limit the interpretation of clinical severity and prognosis. This missing information reduces the completeness of the dataset and may affect the overall evaluation of safety signals.

Seventh, although stratified analyses by age, sex, and time to onset were performed, these results may partly reflect differential reporting behaviors rather than true pharmacological effects. Therefore, the observed demographic or temporal trends should be interpreted with caution and validated in datasets containing more detailed clinical information. Moreover, most reports originated from the United States and Europe, with limited representation from Asia and other regions, which may affect the generalizability of the results across diverse populations.

Eighth, the dataset was dominated by reports from the United States and Europe, whereas data from Asia and other regions were minimal. This geographical imbalance may lead to underrepresentation of regional prescribing practices and population specific characteristics, thereby limiting the global generalizability of the findings. Future studies should aim to include data from multiple pharmacovigilance systems to enhance external validity.

Ninth, because the analysis involved evaluating a large number of Preferred Terms (PTs), the study inherently includes multiple statistical comparisons. Although formal multiplicity corrections are not routinely applied in FAERS based pharmacovigilance research, this may increase the likelihood of false positive signals; therefore, such findings should be interpreted with caution.

Despite these limitations, this large scale real world analysis provides meaningful post marketing insights into the safety profile of fondaparinux and highlights areas requiring further clinical and mechanistic investigation.

## 5. Conclusion

This study conducted a comprehensive analysis of adverse drug events related to fondaparinux using the FAERS database and identified several safety signals not currently described in existing literature. Observed events such as eosinophilia and thrombocytosis may indicate potential immune mediated or hematopoietic effects of fondaparinux; however, these findings are exploratory and require further validation. Although the number of reports for certain signals, such as abnormal coagulation factor X concentration, was small, these observations warrant continued pharmacovigilance and follow up investigations.

By integrating large scale pharmacovigilance data with mechanistic considerations, this study provides an exploratory overview of the potential safety profile of fondaparinux. Nonetheless, inherent limitations of spontaneous reporting systems such as under reporting, incomplete data, and reporting bias should be acknowledged. The predominance of reports from Western countries and the small number of certain adverse events also restrict generalizability. Future multicenter, prospective studies are needed to validate these findings and clarify their clinical relevance.

## Supporting information

**S1 File. Methodological details for disproportionality analyses, including contingency table framework and formulas for ROR, PRR, BCPNN, and EBGM.**
(DOCX)

**S2 File. Complete disproportionality results for fondaparinux associated Preferred Terms (PTs) in the FAERS database.**
(XLSX)

**S3 File. Comparative disproportionality analysis of major bleeding and HIT related adverse events across anticoagulant classes.**
(DOCX)

**S4 File. Age stratified disproportionality results for fondaparinux associated Preferred Terms (PTs) in FAERS.**
(XLSX)

**S5 File. Demographic characteristics, geographic distribution, comorbidities, and concomitant medication profiles of fondaparinux associated FAERS reports.**
(XLSX)

## Author contributions

**Conceptualization:** Yin Wang, Fengqun Cai, Liuyun Wu.

**Data curation:** Fengjiao Kang, Fengqun Cai, Liuyun Wu.

**Formal analysis:** Yin Wang.

**Project administration:** Lizhu Han.

**Supervision:** Qinan Yin, Yuan Bian.

**Writing – original draft:** Fengjiao Kang.

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
