## [Decision Letter · Decision Letter 0]

8 Oct 2025

Dear Dr. Bian,

Thank you for submitting your manuscript to PLOS ONE. After careful consideration, we feel that it has merit but does not fully meet PLOS ONE’s publication criteria as it currently stands. Therefore, we invite you to submit a revised version of the manuscript that addresses the points raised during the review process.

We look forward to receiving your revised manuscript.

Kind regards,

Ignatius Ivan, M.D

Academic Editor

PLOS ONE

Journal Requirements:

National Key Research and Development Program of China (2020YFC2005500)

NO authors have competing interests

5. Thank you for uploading your study's underlying data set. Unfortunately, the repository you have noted in your Data Availability statement does not qualify as an acceptable data repository according to PLOS's standards.

6. Please remove your figures from within your manuscript file, leaving only the individual TIFF/EPS image files, uploaded separately. These will be automatically included in the reviewers’ PDF.

Additional Editor Comments:

1. Methodological Clarity and Reproducibility

Issue: The methods section lacks clarity on the specific R scripts or packages used for data extraction, cleaning, and analysis (e.g., version numbers for openFDA API, R packages for BCPNN/EBGM computation).

Revision Suggestion: Provide detailed methodological transparency by specifying the analytical workflow (R functions or packages), data filters applied, and thresholds for signal detection. This will enhance reproducibility and compliance with pharmacovigilance reporting standards such as the CIOMS VIII guidelines.

2. Lack of Validation Against Comparator Drugs or Negative Controls

Issue: The study focuses solely on Fondaparinux without contextual comparison to similar anticoagulants (e.g., enoxaparin, dalteparin, rivaroxaban). This limits interpretability of the disproportionality signals.

Revision Suggestion: Consider adding a sensitivity analysis or discussion comparing signal magnitude and pattern to a pharmacologically similar agent. This would strengthen claims about the distinct safety profile of Fondaparinux.

3. Overinterpretation of Signal Strength (Causality vs. Association)

Issue: The discussion implies causal relationships (“Fondaparinux may induce HIT” or “should be included in the drug label”), which overstates the inferential capacity of FAERS data.

Revision Suggestion: Revise such statements to emphasize signal detection and hypothesis generation, not causation. A clearer distinction between statistical signal and clinical causality is needed throughout the discussion and conclusion.

4. Limited Consideration of Reporting Bias and Confounding

Issue: Although the paper mentions underreporting and bias briefly, it does not quantify or adjust for these biases. For instance, the large number of reports from consumers versus clinicians (45.91%) may distort clinical signal interpretation.

Revision Suggestion: Expand the Limitations section to discuss confounding by indication, stimulated reporting (e.g., post-ACS approval in 2007), and missing denominator data. Where possible, suggest statistical corrections or triangulation with other data sources (e.g., VigiBase, EudraVigilance).

5. Data Presentation and Consistency Issues

Issue: Figure 2 caption incorrectly refers to “Ambrisentan” instead of “Fondaparinux.” Inconsistent capitalization and redundant labeling also appear in tables and figures.

Revision Suggestion: Conduct thorough proofreading to correct labeling errors and ensure consistency between text, tables, and figures (e.g., Table 2 SOC classification titles, Figure 3 panel descriptions). Ensure all figures are clearly linked to the text narrative and include legends that are self-explanatory.

6. Clinical Interpretation of Rare Events

Issue: The discussion highlights rare events such as abnormal coagulation factor X concentration (n=3) and antiphospholipid syndrome (n=6) as potential novel signals but lacks adequate caution regarding the reliability of such small counts.

Revision Suggestion: Temper conclusions on rare AEs and explicitly discuss false-positive risks due to low event frequency and multiple testing. Emphasize the exploratory nature and the need for prospective validation.

7. Incomplete Integration with Existing Literature and Regulatory Implications

Issue: The discussion is largely descriptive and lacks engagement with contemporary pharmacovigilance literature or prior disproportionality analyses involving Fondaparinux or factor Xa inhibitors.

Revision Suggestion: Strengthen the discussion by comparing current findings with existing post-marketing surveillance studies and FDA/EMA safety updates. Include references on recent data-mining frameworks or label-change precedents to contextualize clinical and regulatory implications.

Reviewers' comments:

Reviewer's Responses to Questions

**Comments to the Author**

1. Is the manuscript technically sound, and do the data support the conclusions?

Reviewer #1: Yes

Reviewer #2: Partly

Reviewer #3: Yes

2. Has the statistical analysis been performed appropriately and rigorously?

Reviewer #1: I Don't Know

Reviewer #2: No

Reviewer #3: Yes

3. Have the authors made all data underlying the findings in their manuscript fully available?

Reviewer #1: Yes

Reviewer #2: No

Reviewer #3: Yes

4. Is the manuscript presented in an intelligible fashion and written in standard English?

Reviewer #1: Yes

Reviewer #2: Yes

Reviewer #3: Yes

Reviewer #1: This is a disproportionality analysis of the FAERS to identify potential safety signals of fondaparinux. The paper is clear and well-written. Please see below for a few suggestions.

1. The Abstract contains many abbreviations that are not defined (e.g., FAERS, ROR, PRR, BCPNN, EBGM). Ideally these should be defined with first use in both the Abstract and main text.

2. Figure 2 – The incorrect drug (ambrisentan) is listed in the figure title. Please ensure that the data apply to fondaparinux and not to ambrisentan.

3. Fondaparinux is capitalized throughout the manuscript. It should not be capitalized as it is a generic name.

4. Table 1 – For age, please specify what is being shown? Is it a mean or median? Is the range an IQR? Please also define “tto” and “ttoQ” in the table or table legend.

5. The authors write, “…most adverse events in both male and female patients occurred within the first month of treatment…” Would it be more accurate to say, “…within the first month of INITIATING treatment…”?

6. The authors write, “However, it remains necessary to continue safety monitoring throughout the entire treatment period and during the extended follow -up phase, potentially up to one year after drug discontinuation.” I disagree with this sentence. As one would expect, the authors identified bleeding as the main toxicity of fondaparinux. The half-life of fondaparinux is about 18 hours. Thus it is unrealistic to think that fondaparinux could contribute to bleeding more than a few days after stopping. I cannot think of any toxicities, identified by the authors, that could be expected to occur as a result of fondarinux months after it was stopped.

7. It is remarkable that AE reporting for fondaparinux has significantly decreased over the last several years. It would be interesting if the authors speculated as to why this might be. Is it because the drug is no longer new and there is thus less incentive to report adverse effects? Or could it be because it is used far less than it once was, possibly due to increased use of direct oral anticoagulants.

8. It is worth noting that hematologists do not recognize a single disorder called “thrombosis and thrombocytopenia syndrome”. Rather, this is a group of disorders associated with both thrombosis and thrombocytopenia including but not limited to heparin-induced thrombocytopenia, other PF4 disorders such as vaccine-induced thrombotic thrombocytopenia, antiphospholipid syndrome, disseminated intravascular coagulation, and thrombotic thrombocytopenic purpura.

Reviewer #2: The study addresses an important pharmacovigilance question on the safety profile of fondaparinux using FAERS data and multiple disproportionality algorithms (ROR, PRR, BCPNN, EBGM). The dataset is large, spanning 10 years, and the analysis identifies both known and novel safety signals. However, there are significant methodological, interpretational, and presentation issues that limit the scientific rigor and impact.

1. Title of the study should be revised.

2. Introduction is a bit short and avoid redundancy, revise accordingly

3. The manuscript often implies causal links, but FAERS can only generate signals, not establish causality. Stronger disclaimers are needed throughout, revise accordingly for better clarity and impact

4. FAERS reports are prone to underreporting, duplication, missing covariates, and confounding. Although limitations are mentioned, their implications for interpretation are underplayed. Discussion all the issues accordingly throughout the revised manuscript.

5. Cite reference for FAERS in method part.

6. Also cite relevant references in the method part i.e. Medex_UIMA_1.8.3 system, Preferred Terms, (PTs) and System Organ Classes (SOCs) from MedDRA, Reporting Odds Ratio (ROR), Proportional Reporting Ratio (PRR), Bayesian Confidence Propagation Neural Network (BCPNN), and Empirical Bayesian Geometric Mean (EBGM) (20-23)

7. The chosen thresholds for disproportionality signals are not fully justified or compared with established best practices. Adjustments for multiple testing are not discussed.

8. Without exposure data (number of patients on fondaparinux), the relative risk cannot be estimated. This limitation should be emphasized more. This issue may cause negative impact on the generalizability of results.

9. Some signals are based on extremely few cases (e.g., abnormal factor X concentration, n=3 ), yet interpreted as meaningful. Such findings lack statistical power and clinical reliability.

10. While interesting, stratified results (age, gender, onset) may reflect reporting bias rather than true pharmacological effects. This caveat is insufficiently acknowledged.

11. The discussion part does not adequately distinguish between well-established adverse events (e.g., bleeding, hematoma) and weak, hypothesis-generating signals.

12. Several figures are descriptive but not critically analyzed. For example, time-to-onset curves are shown but not contextualized clinically.

13. Captions and legends of figures and tables should be standalone and self-explanatory, revise accordingly.

14. Table formatting is not good for publication, revise for better clarity, readership and impact.

15. Claims about “new” adverse events would benefit from a systematic side-by-side comparison with FDA/EMA product labelling.

16. Statements such as recommending monitoring of anti-PF4 antibodies and eosinophils are premature, given the weak evidence base. This is overinterpretation. Revise accordingly.

17. The dataset is dominated by US and European cases, with minimal Asian data . This limitation is noted but should be emphasized earlier.

18. The conclusion suggests changes to labeling and practice, but the evidence presented does not justify such strong recommendations.

19. Several grammatical issues and long sentences reduce readability.

20. The terminology “Adverse drug events” vs. “adverse drug reactions” are used interchangeably—should be standardized.

21. About 30% of reports lacked outcome data , but this important limitation is buried rather than highlighted.

22. Ethical approval required or provide a statement clarifying that ethical approval was not applicable.

Reviewer #3: Hematoma/hemorrhage/hematoma are expected in surgical patients receiving anticoagulants. The FAERS analysis must try to separate background risk (e.g., post-operative bleeding) from drug-specific signal.

At minimum, report how many bleeding/hematoma reports included concomitant heparin/LMWH or warfarin or antiplatelet agents. How many HIT-like and TTS reports had recent/ concurrent heparin exposure? Provide stratified results.

For the HIT-like signals and thrombosis-with-thrombocytopenia, present how many reports had recent heparin exposure or concurrent heparin/LMWH. Without this, attributing HIT to fondaparinux is prematureWhile the study identifies specific adverse events, the small number of reports for certain signals (e.g., abnormal concentrations of coagulation factor X) raises concerns regarding the statistical power and clinical significance of these findings.

**Do you want your identity to be public for this peer review?** For information about this choice, including consent withdrawal, please see our Privacy Policy

Reviewer #1: No

Reviewer #2: No

Reviewer #3: **Yes:** Dr Shabana Ali

---

## [Author Response · Author response to Decision Letter 1]

20 Nov 2025

Letter to the Editor and Reviewers

Dear Editor and Reviewers,

We would like to express our sincere gratitude for your careful review and constructive feedback on our manuscript. We truly appreciate the time and effort that the editor and the three reviewers have devoted to providing detailed and insightful comments, which have been instrumental in improving the scientific rigor, logical clarity, and international quality of our work.

After receiving the comments, we thoroughly analyzed and addressed each point. The revised manuscript has been substantially improved in research design, data interpretation, and academic presentation. The main revisions and improvements are summarized as follows:

Title and Abstract

The title has been refined for academic precision. All abbreviations (FAERS, ROR, PRR, BCPNN, EBGM) are now defined upon first use in both the Abstract and the main text to enhance clarity.

Methodological Transparency

The FAERS database citation was added, and detailed descriptions of data extraction and cleaning were included. The R packages (with version numbers) used for disproportionality analysis are now specified. Section 2.3 clarifies the rationale for signal detection thresholds with appropriate literature support, ensuring reproducibility and compliance with CIOMS VIII guidelines.

Results and Stratified Analyses

The results section was reorganized for logical consistency. Additional stratified analyses were conducted for reports involving concomitant antiplatelet or anticoagulant exposure to address concerns regarding background surgical bleeding risk.

Discussion Revision

The discussion was extensively restructured to distinguish established anticoagulant related adverse events (e.g., hemorrhage, hematoma) from exploratory or hypothesis generating signals (e.g., HIT like reactions, abnormal factor X concentrations). Overinterpretations were removed, and clearer explanations were added regarding the limitations of FAERS based inference.

Expanded Limitations Section

A dedicated subsection has been expanded to emphasize FAERS limitations, including underreporting, duplication, missing covariates, lack of exposure denominators, reporting bias, and geographic imbalance (predominantly U.S. and European data). Approximately 30% of reports lacked outcome data, which is now explicitly highlighted.

Figures, Tables, and Language Editing

Figure legends and table captions were revised to be self explanatory. Table formats were standardized, and the incorrect drug name in Figure 2 was corrected. The generic name “fondaparinux” is used consistently, and all grammatical and stylistic issues were carefully revised to improve readability.

Ethical Statement and Conclusion

An ethical statement has been added, clarifying that ethical approval was not required because FAERS is a publicly available, de-identified database. The conclusion was rewritten to remove unsupported recommendations, ensuring that the findings are presented cautiously and empirically.

Funder and Competing Interests

We have updated both the Role of Funder and Competing Interests statements as requested, including the required sentence in full.

We believe that these substantial revisions have greatly enhanced the scientific quality, transparency, and overall coherence of the paper. We sincerely thank the editor and reviewers for their detailed and constructive comments, which have guided us in refining the manuscript to its current improved version.

Editor comment 1: Methodological Clarity and Reproducibility

Issue: The methods section lacks clarity on specific R scripts or packages used for data extraction, cleaning, and analysis (e.g., version numbers for openFDA API, R packages for BCPNN/EBGM computation).

Response:

We greatly appreciate the editor’s constructive comment. In the revised manuscript, we have substantially enhanced the transparency and reproducibility of the analytical methods.

Sections 2.2 through 2.6 now provide a detailed description of the entire workflow, including data extraction, cleaning, MedDRA coding, and signal detection. The R version (4.3.0) and all major packages used (dplyr, data.table, epiR, DescTools, bayesAB, openEBGM, and ggplot2) are explicitly listed. The threshold criteria for each algorithm are specified as follows: ROR (lower 95% CI > 1), PRR (≥ 2 with χ² ≥ 4), BCPNN (IC025 > 0), and EBGM (EB05 > 1).

We also clarified that all analyses were executed using fully scripted workflows in R Studio and cross-verified in Microsoft Excel for accuracy. These revisions align with the CIOMS VIII recommendations and ensure full methodological transparency and reproducibility of our study.

Editor comment 2: Lack of Validation Against Comparator Drugs or Negative Controls

Issue: The study focuses solely on fondaparinux without contextual comparison to similar anticoagulants (e.g., enoxaparin, dalteparin, rivaroxaban), limiting interpretability of the disproportionality signals.

Response:

We thank the editor for this valuable and insightful recommendation. In the revised manuscript, we have added and expanded comparative and sensitivity analyses, as presented in Section 3.2.3 “Stratified and comparative analysis across anticoagulant exposures.”

First, a stratified analysis was performed to compare the disproportionality of major adverse events in fondaparinux users with and without concomitant anticoagulant or antiplatelet therapy. The significant bleeding signals (hematoma, muscle haemorrhage, and haemorrhage) persisted even after excluding concomitant medications, indicating that these associations were largely independent of polypharmacy.

Second, a comparative disproportionality analysis was conducted across other major anticoagulant classes, including enoxaparin, unfractionated heparin (UFH), and direct oral anticoagulants (DOACs: apixaban, dabigatran, edoxaban, and rivaroxaban), using identical algorithms (ROR, PRR, BCPNN, EBGM) and detection thresholds. The results showed consistent positive bleeding signals across all anticoagulants, confirming a class- wide effect, whereas HIT related signals were observed only for fondaparinux and UFH, suggesting drug specific associations.

These additions have been incorporated into both the Results and Discussion sections, supported by relevant literature and regulatory references, thereby strengthening the interpretability and robustness of our findings.

Editor comment 3: Overinterpretation of Signal Strength (Causality vs. Association)

Issue: The Discussion implies causal relationships (e.g., “fondaparinux may induce HIT” or label change suggestions), which exceeds what FAERS data can support.

Response:

We sincerely appreciate the editor’s valuable observation. The Discussion and Conclusion have been thoroughly reviewed and revised to avoid implying causality:

All potentially causal expressions were rewritten into statistical association or signal based language—for example, “may induce” was replaced with “was associated with HIT like reactions,” and these were explicitly defined as case report level and hypothesis generating findings;

Sentences suggesting label changes or recommendations beyond the strength of evidence were removed;

We added explicit statements at the beginning of Section 4 (Discussion) and in Section 4.5 (Limitations) clarifying that FAERS is intended for disproportionality signal detection and hypothesis generation, not causal inference;

Phrases that might imply causality (e.g., “likely pharmacologically driven,” “underscore the importance of… monitoring”) were replaced with more cautious wording such as “may reflect pharmacologic plausibility” and “support prioritizing monitoring as an exploratory signal”;

In the TTO (Section 4.4) and Rare events (Section 4.3) subsections, we included explicit cautionary statements such as “should be interpreted cautiously” and “should be regarded as exploratory and hypothesis generating,” emphasizing that these observations are exploratory and warrant further validation.

These revisions ensure that all interpretations are scientifically cautious, aligned with the evidentiary boundaries of FAERS data, and consistent across the manuscript.

Editor comment 4: Limited Consideration of Reporting Bias and Confounding

Issue: Although reporting bias is briefly mentioned, the manuscript does not adequately address consumer reporting proportion, indication confounding, or the lack of exposure denominators.

Response:

We appreciate the editor’s valuable feedback. The Limitations section (Section 4.5) has been expanded to provide a more thorough discussion of potential reporting bias and confounding factors. The key additions include:

Reporting source imbalance: Approximately 45.91% of reports were submitted by consumers, which may affect data accuracy and clinical interpretation;

Indication confounding and background risk: Because fondaparinux is frequently administered for postoperative thromboprophylaxis, bleeding events may be partly attributable to surgical factors rather than drug specific effects. We have added a statement acknowledging that residual confounding may remain even after excluding concomitant anticoagulants or antiplatelet agents;

Lack of exposure denominators: We noted that FAERS lacks patient level exposure data, making it impossible to calculate absolute incidence or risk; the present findings therefore represent disproportionality signals rather than measures of relative risk;

Geographic imbalance and external validation: The dataset is dominated by U.S. and European reports, with minimal Asian data. We recommend future studies to triangulate findings using multiple pharmacovigilance databases such as VigiBase and EudraVigilance to enhance external validity.

These revisions provide a more comprehensive and balanced acknowledgment of potential reporting bias and confounding within the study limitations.

Editor comment 5: Data Presentation and Consistency Issues

Issue: Figure 2 caption incorrectly refers to “Ambrisentan” instead of “Fondaparinux.” Inconsistent capitalization and redundant labeling also appear in tables and figures.

Response:

We sincerely thank the editor for this meticulous observation. All figures and tables have been carefully proofread and revised for consistency:

Correction of figure titles: The caption of Figure 2 has been corrected from “Ambrisentan” to “Fondaparinux,” and all figure titles have been verified for accuracy;

Terminology and formatting standardization: Abbreviations such as SOC (System Organ Class), PT (Preferred Term), and TTO (Time to Onset) have been consistently defined and explained within the figure legends;

Improved self explanatory legends: Each figure and table (Figures 1–5, Tables 1–4) now includes a complete and standalone legend, ensuring they can be interpreted independently of the main text;

Text–figure alignment: Cross references in the text (e.g., “Figure 3 visualizes …,” “Figure 4 illustrates …”) were checked for full correspondence with figure content.

These revisions enhance the accuracy, readability, and overall presentation quality of the manuscript.

Editor comment 6: Clinical Interpretation of Rare Events

Issue: The discussion highlights rare events such as abnormal coagulation factor X concentration (n = 3) and antiphospholipid syndrome (n = 6) as potential novel signals but lacks adequate caution regarding the reliability of such small counts.

Response:

We thank the editor for this thoughtful and important comment. The relevant text in Section 4.3 “Rare reports” has been carefully revised to ensure cautious interpretation of rare signals.

All findings based on very limited numbers of cases (e.g., abnormal coagulation factor X concentration, thrombosis with thrombocytopenia, antiphospholipid syndrome, eosinophilia, thrombocytosis) are now explicitly described as exploratory or hypothesis generating.

We added qualifying statements such as “should be interpreted cautiously” and “should be regarded as exploratory and hypothesis generating” to emphasize that these results are preliminary and require further validation.

These revisions ensure that the discussion accurately reflects the statistical and exploratory nature of FAERS data while maintaining scientific caution.

Editor comment 7: Incomplete Integration with Existing Literature and Regulatory Implications

Issue: The discussion lacks engagement with prior pharmacovigilance studies and regulatory updates from FDA or EMA, limiting the contextual and regulatory interpretation of the findings.

Response:

We appreciate the editor’s valuable recommendation. In the revised manuscript, we have strengthened the integration with existing literature and regulatory information, particularly in Section 4.2 and the closing paragraphs of the Discussion.

Comparison with prior studies: Additional discussion was added comparing our findings with previous pharmacovigilance analyses (e.g., Frontiers in Oncology, 2023), showing that the identified hematologic and bleeding signals for fondaparinux are consistent with earlier postmarketing observations;

Inclusion of regulatory information: References to the 2024 FDA prescribing information and the 2024 EMA EPAR (Product Information for Arixtra) were incorporated to highlight that both agencies list haemorrhage and thrombocytopenia in fondaparinux’s safety labeling and recommend renal function monitoring;

Class level context: We noted the alignment between fondaparinux and other factor Xa inhibitors in terms of safety profile and regulatory labeling, reinforcing the external validity of our findings;

Avoidance of overinterpretation: Statements proposing label modification or clinical changes were removed, ensuring that our conclusions focus on pharmacovigilance observation rather than regulatory action.

These revisions enhance the manuscript’s scholarly depth, align it with current regulatory knowledge, and strengthen the contextual interpretation of the findings.

Reviewer 1 – Comment 1

Comment:

The Abstract contains many abbreviations that are not defined (e.g., FAERS, ROR, PRR, BCPNN, EBGM). Ideally these should be defined with first use in both the Abstract and main text.

Response:

We thank the reviewer for this helpful suggestion. In the revised manuscript, all abbreviations have been clearly defined at their first appearance in both the Abstract and the main text. Specifically, the following definitions were added: FAERS = FDA Adverse Event Reporting System; ROR = Reporting Odds Ratio; PRR = Proportional Reporting Ratio; BCPNN = Bayesian Confidence Propagation Neural Network; and EBGM = Empirical Bayes Geometric Mean. These clarifications improve the readability and precision of the manuscript.

Reviewer 1 – Comment 2

Comment:

Figure 2 – The incorrect drug (ambrisentan) is listed in the figure title. Please ensure that the data apply to fondaparinux and not to ambrisentan.

Response:

We appreciate the reviewer’s careful observation. All data have been verified to correspond exclusively to fondaparinux. The caption of Figure 2 has been corrected from “Ambrisentan” to “Fondaparinux,” and the figure legend and corresponding text references have been updated accordingly.

Reviewer 1 – Comment 3

Comment:

Fondaparinux is capitalized throughout the manuscript. It should not be capitalized as it is a generic name.

Response:

Thank you for noting this. All occurrences of “Fondaparinux” have been changed to lowercase “fondaparinux,” and capitalization has been standardized for all generic drug names throughout the manuscript.

Reviewer 1 – Comment 4

Comment:

Table 1 – For age, please specify what is being shown? Is it a mean or median? Is the range an IQR? Please also define “tto” and “ttoQ” in the table or table legend.

Response:

We appreciate this helpful comment. In the revised Table 1, the data presentation has been clarified. Continuous variables such as Age (years) and Time to onset (days) are now explicitly expressed as median (interquartile range, IQR), as stated in the table footnote. The previous abbreviations “TTO” and “TTOQ” have been removed to improve readability and ensure consistency across the manuscript.

Reviewer 1 — Comment 5

Comment �

“The authors write, ‘…most adverse events in both male and female

---

## [Decision Letter · Decision Letter 1]

15 Dec 2025

Dear Dr. Bian,

Thank you for submitting your manuscript to PLOS ONE. After careful consideration, we feel that it has merit but does not fully meet PLOS ONE’s publication criteria as it currently stands. Therefore, we invite you to submit a revised version of the manuscript that addresses the points raised during the review process.

We look forward to receiving your revised manuscript.

Kind regards,

Ignatius Ivan, M.D

Academic Editor

PLOS One

Journal Requirements:

Reviewers' comments:

Reviewer's Responses to Questions

**Comments to the Author**

Reviewer #1: (No Response)

Reviewer #2: All comments have been addressed

Reviewer #3: All comments have been addressed

2. Is the manuscript technically sound, and do the data support the conclusions?

Reviewer #1: Yes

Reviewer #2: Yes

Reviewer #3: Yes

3. Has the statistical analysis been performed appropriately and rigorously?

Reviewer #1: Yes

Reviewer #2: Yes

Reviewer #3: Yes

4. Have the authors made all data underlying the findings in their manuscript fully available?

Reviewer #1: Yes

Reviewer #2: Yes

Reviewer #3: Yes

5. Is the manuscript presented in an intelligible fashion and written in standard English?

Reviewer #1: Yes

Reviewer #2: No

Reviewer #3: Yes

Reviewer #1: Thank you to the authors for their thoughtful responses and revisions. I have only minor suggestions remaining:

1. Abstract – Define PT with first use

2. Results, “The age distribution of reporters showed that the majority were over 65 years old…” – I assume you mean the age distribution of patients on which reports were filed (not the age distribution of reporters).

3. Thank you for de-capitalizing fondaparinux. However, there are some instances where “fondaparinux” is the first word in a sentence and should therefore be capitalized.

Reviewer #2: I have no further suggestions for improvement. The manuscript is now improved. If the other reviewers also agrees, the manuscript can be considered for publication

Best Regards.

Reviewer #3: The authors have addressed almost all the queries raised by the reviewer. There are none of the queries remaining unanswered.

**Do you want your identity to be public for this peer review?** For information about this choice, including consent withdrawal, please see our Privacy Policy

Reviewer #1: No

Reviewer #2: No

Reviewer #3: **Yes:** Prof Dr Shabana Ali

---

## [Author Response · Author response to Decision Letter 2]

16 Dec 2025

We thank the Academic Editor and all reviewers for their careful evaluation and constructive comments. We have revised the manuscript accordingly. Specifically, we have defined Preferred Term (PT) at its first occurrence in the Abstract, corrected the wording in the Results section to clarify that the age distribution refers to patients rather than reporters, and ensured that “Fondaparinux” is capitalized when it appears at the beginning of a sentence. All comments raised by the reviewers have been fully addressed, and the manuscript has been carefully revised for clarity and consistency.

---

## [Editor Report · Decision Letter 2]

26 Jan 2026

Disproportionality Analysis of Fondaparinux Associated Adverse Events Based on the FDA Adverse Event Reporting System

PONE-D-25-40001R2

Dear Dr. Yuan Bian

We’re pleased to inform you that your manuscript has been judged scientifically suitable for publication and will be formally accepted for publication once it meets all outstanding technical requirements.

Kind regards,

Ignatius Ivan, M.D

Academic Editor

PLOS One
---

## [Editor Report · Acceptance letter]

PONE-D-25-40001R2

PLOS One

Dear Dr. Bian,

I'm pleased to inform you that your manuscript has been deemed suitable for publication in PLOS One. Congratulations! Your manuscript is now being handed over to our production team.

Kind regards,

on behalf of

dr. Ignatius Ivan

Academic Editor

PLOS One